# Whole Transcriptome Analyses of Apricots and Japanese Plum Fruits after 1-MCP (Ethylene-Inhibitor) and Ethrel (Ethylene-Precursor) Treatments Reveal New Insights into the Physiology of the Ripening Process

**DOI:** 10.3390/ijms231911045

**Published:** 2022-09-20

**Authors:** Juan A. Salazar, David Ruiz, Patricio Zapata, Pedro J. Martínez-García, Pedro Martínez-Gómez

**Affiliations:** 1Department of Plant Breeding, CEBAS-CSIC, Espinardo, 30100 Murcia, Spain; 2Facultad de Medicina Y Ciencia, Universidad San Sebastián, Santiago 7510157, Chile

**Keywords:** *Prunus*, apricot, Japanese plum, fruit quality, ripening, transcriptomics, de novo sequencing, ethylene-related phytohormones

## Abstract

The physiology of *Prunus* fruit ripening is a complex and not completely understood process. To improve this knowledge, postharvest behavior during the shelf-life period at the transcriptomic level has been studied using high-throughput sequencing analysis (RNA-Seq). Monitoring of fruits has been analyzed after different ethylene regulator treatments, including 1-MCP (ethylene-inhibitor) and Ethrel (ethylene-precursor) in two contrasting selected apricot (*Prunus armeniaca* L.) and Japanese plum (*P. salicina* L.) cultivars, ‘Goldrich’ and ‘Santa Rosa’. KEEG and protein–protein interaction network analysis unveiled that the most significant metabolic pathways involved in the ripening process were photosynthesis and plant hormone signal transduction. In addition, previously discovered genes linked to fruit ripening, such as pectinesterase or auxin-responsive protein, have been confirmed as the main genes involved in this process. Genes encoding pectinesterase in the pentose and glucuronate interconversions pathway were the most overexpressed in both species, being upregulated by Ethrel. On the other hand, auxin-responsive protein IAA and aquaporin PIP were both upregulated by 1-MCP in ‘Goldrich’ and ‘Santa Rosa’, respectively. Results also showed the upregulation of chitinase and glutaredoxin 3 after Ethrel treatment in ‘Goldrich’ and ‘Santa Rosa’, respectively, while photosystem I subunit V psaG (photosynthesis) was upregulated after 1-MCP in both species. Furthermore, the overexpression of genes encoding GDP-L-galactose and ferredoxin in the ascorbate and aldarate metabolism and photosynthesis pathways caused by 1-MCP favored antioxidant activity and therefore slowed down the fruit senescence process.

## 1. Introduction

Fruit species can be classified as climacteric or non-climacteric, depending on whether they show an increase in respiration rate during fruit ripening or not. Therefore, we consider climacteric fruit those species which experience a respiration rate increase which is usually linked with increased ethylene emission. The Rosaceae family houses a variety of climacteric fruit species, including *Prunus* species such as peach (*Prunus persica* L.), apricot (*Prunus armeniaca* L.), or Japanese plum (*Prunus salicina* L.), all of them being characterized by their diploid genome (2n = 16) and their limited postharvest life. Apricot does not usually exceed 2 weeks of cold storage at 0 °C and 90% of humidity [1], while peach could reach between 3 or 4 weeks of cold storage [2], and plum could overcome 4 weeks [3]. Therefore, the export fitness of stone fruit depends directly on its shelf-life period. In addition, the study of the changes that take place in the fruit ripening process at the biochemical, metabolic, and molecular levels to know the key factors that could favor the increase in the shelf-life period of the new cultivars in the current apricot and plum breeding programs must be studied. Thus, new studies and conservation techniques should be addressed in apricots and Japanese plums, as reported in peach cultivars, considering the *Prunus* model species [4,5,6].

Fruit ripening involves a set of processes that starts from the last stages of growth and development that define the characteristics in the aesthetics and the quality, which are mainly associated with color, texture, and sensory attributes such as flavor. From a physiological point of view, ripening is a genetically programmed process that includes important physical, chemical, and biochemical changes that make the fruit more appealing to the consumer. During the fruit ripening process, biochemical events that induce color changes occur, associated with chlorophyll breakdown or the synthesis and accumulation of other pigments such as carotenoids and anthocyanins [7,8,9], changes in texture that result in the fruit softening due to the solubilization of pectin, degradation of starch, and changes in protein content and hydration of the cell wall [10], and variations in taste and aroma due to the accumulation of sugars and organic acids and the production of volatiles as esters and alcohols [11,12].

At the transcriptomic level, ethylene and auxin-related genes, which are undoubtedly linked to fruit ripening [13], are involved in oxidative stress in peach. Fruit ripening also includes taste and aroma changes affecting organoleptic qualities. In recent works, new sugar transporters and transcription factors linked to the transport of sugars, organic acid, and volatiles have been identified in apricot, playing an important role in flavor quality development [14]. Recently, after apricot transcriptome analysis, new differentially expressed genes linked to the flavonoid, starch and sucrose, and carotenoid metabolism have been identified, suggesting a possible design of molecular markers useful for assisted selection [15]. Fruit ripening also induces important metabolic changes in the different pathways, which are associated with the increase in respiration rate and synthesis of ethylene [16] or changes in the metabolism of starch and organic acids [17]. Additionally, at the molecular level, the fruit ripening process involves changes in gene expression through the synthesis of ripening-specific mRNAs, interference with RNAs, appearance and disappearance of mRNAs [14,18], and variations in protein expression by the synthesis of de novo ripening-specific proteins or disappearance of proteins [19]. In addition, quantitative proteomic analysis in peach ripening by iTRAQ has also revealed that NCED is a key protein in ABA synthesis favoring respiration and fruit softening, while a negative correlation between polygalacturonase/pectase lyase and fruit softening was shown [20].

On the other hand, the role of ethylene hormone in fruit ripening has been extensively studied for decades, especially in the last few years, where more current works have identified new components in the ethylene signal pathway, which are ethylene response factors (ERFs) [21]. The process can be divided into three stages: (1) ethylene biosynthesis, initiating with S-adenosyl-Met and comprising two steps catalized by 1-aminocyclopropane-1-carboxylic acid synthase (ACS) and 1-aminocyclopropane-1-carboxylic acid oxidase (ACO), the last one being the key of the conversion of ACC to ethylene; (2) the ethylene signal pathway, where the presence of ethylene is captured in the endoplasmic reticulum (ER) by ethylene protein receptors (ETR), with Ethylene-insensitive protein 2 (EIN2) being considered a key signal transducer activating the expression of target genes; (3) the ethylene response, where a transcriptional cascade is initiated involving EIN3/EIL1 and ERFs which regulate ripening-related traits [22]. For this reason, chemical ethylene regulators such as 1-MCP and etephon (Ethrel^®^) have been widely used in order to regulate the ripening of climacteric fruits. 1-methylcyclopropene (1-MCP) is a molecule which adheres to ethylene receptors (ER) in the endoplasmic reticulum, inhibiting ethylene emission and favoring a longer fruit shelf-life period [23], while Ethrel accelerates the onset of ethylene emission and hence activates genes related to ethylene biosynthesis, such as ACS and ACO [24].

The main goal of this study was to dissect, at the transcriptomic level, the response triggered by the application of ethylene-related regulators in apricot (*Prunus armeniaca* L.) and Japanese plum (*P. salicina* L.) fruits to display an entire picture of the fruit ripening mechanism by evaluating key related genes and proteins through the analysis of metabolic pathways and protein−protein interaction networks.

## 2. Results

### 2.1. Monitoring of Fruit Quality and Postharvest Traits

Fruit quality and postharvest traits were evaluated in apricot cultivars ‘Moniquí’, ‘Goldrich’, and ‘Dorada’, and Japanese plum cultivars ‘Golden Japan’, ‘Santa Rosa’, ‘Black Splendor’, and ‘Angeleno’ (Table 1) after 1-MCP and Ethrel chemical treatments.

In the case of apricot cultivars (Appendix A; Table 2), at harvest time, ‘Goldrich’ showed the greatest fruit weight and a more orange skin color (h° = 77.5) than ‘Moniquí’, which showed light yellow skin color (h° = 95.4), while ‘Dorada’ displayed similar skin color to ‘Goldrich’ (Table 2 and Appendix A). On the other hand, ‘Goldrich’ showed the lowest SSC/Malic ratio (4.05) due to its very high acidity level (>3 g 100 mL^−1^), while ‘Dorada’ reached the highest SSC/Malic ratio (12.42). Regarding SSC, ‘Moniquí’ displayed the highest SSC value (14.5%) and intermediate acidity level (1.64 g 100 mL^−1^) (Table 2 and Appendix A). Furthermore, ‘Moniquí’ experienced the biggest skin and flesh color change during the first four days, decreasing the h° value from 95° to 85° towards more orange tones (Appendix A); however, no differences between treatments were shown. On the other hand, ‘Goldrich’ and ‘Dorada’ cultivars displayed differences between days and treatments in terms of fruit color (Appendix A). Overall, SSC and malic acid experienced a slight increase and decrease, respectively, during the postharvest period. However, no differences were found either between treatments or over time, except in ‘Goldrich’, which exhibited differences between treatments on its last day (Appendix A). 

In the case of fruit firmness and ethylene emission, ‘Goldrich’ was the most contrasting cultivar in terms of firmness on day 4 (Figure 1), while ‘Moniquí’ and ‘Dorada’ showed lower differences between treatments and days. However, ‘Moniquí’ showed greater significant differences in terms of ethylene emission but not in fruit firmness. With this in mind, we decided to select the ‘Goldrich’ cultivar for the following analyses because this genotype showed significant differences in both firmness and ethylene emission, these being the most relevant traits in terms of postharvest behavior.

Regarding Japanese plum cultivars, ‘Black Splendor’ and ‘Angeleno’ recorded the highest fruit weight, around 70 and 80 g (Table 3). However, ‘Santa Rosa’ experienced the most significant differences between treatments and days for skin and flesh color. This cultivar’s skin color turned from red to purple (Appendix A). On the other hand, ‘Angeleno’ displayed differences between treatments on the last day of phenotyping (day 24) (Appendix A), with it’s flesh color turning from yellow to red. In terms of fruit firmness and ethylene evolution of the evaluated cultivars, ‘Santa Rosa’ recorded the most significant differences in fruit firmness and ethylene emission, exhibiting a maximum ethylene peak on day 4 (Figure 2). On the other hand, ‘Angeleno’ was the most suppressed-climacteric cultivar, even without 1-MCP application (no ethylene emission) or after Ethrel treatment (low ethylene emission only after 21 days). Therefore, the ‘Santa Rosa’ cultivar was selected for further analysis.

The PCA results (Appendix A) showed that ‘Goldrich’ and ‘Moniquí’ were the most contrasting apricot cultivars, with ‘Moniquí’ being more influenced by ethylene emission, respiration rate, skin, and flesh color (h°), while ‘Goldrich’ experienced major changes in flesh and skin color. As for the plum cultivars, ‘Black splendor’ and ‘Santa Rosa’ experienced major changes in flesh color, respiration rate, ethylene emission, and loss of weight contrary to ‘Angeleno’ and ‘Golden Japan’. In addition, if we analyze the treatments per each apricot cultivar (Appendix A) separately, the most contrasting was ‘Goldrich’, in which 1-MCP and Ethrel showed an opposite effect. Regarding Japanese plum cultivars, ‘Santa Rosa’ and ‘Black Splendor’ showed the most contrasting behavior between treatments, especially ‘Santa Rosa’, reaching the highest explanation in the *x*-axis (Appendix A).

‘Goldrich’ and ‘Santa Rosa’ were displayed as the best candidates in order to gain a deeper understanding of the mechanisms involved in the fruit ripening process at the molecular level. Additionally, chlorophyll and carotenoid content were determined for ‘Goldrich’ and ‘Santa Rosa’ (Appendix A). Regarding the carotenoid content, no differences were found in ‘Goldrich’ nor ‘Santa Rosa’, ranging between 8–10 mg 100 mL^−1^ and between 2–3 mg 100 mL^−1^, respectively. However, the chlorophyll content showed an important breakdown in apricot ‘Goldrich’ and Japanese plum ‘Santa Rosa’, with chlorophyll being totally degraded except for the 1-MCP treatment on day 4 for both species.

### 2.2. De Novo Transcriptome Assembly, Functional Annotation, and Transcripts Classification

In the current work, de novo transcriptome assembly was performed in order to reconstruct the transcript sequences without reference genome sequence in apricot ‘Goldrich’ and Japanese plum ‘Santa Rosa’ on their most contrasting day (day 4). After sequencing, the raw data production showed a Phred quality score greater than or equal to 20 or 30 (Q20 and Q30) over 90% for both species (Appendix A). Therefore, considering that a Phred quality score of 20 means 99% accuracy, reads over a score of 20 were accepted as good quality. As a first overall result of our essay, count reads were analyzed by PCA in order to compare the dispersion of each replicate within each treatment (Appendix A). In the PCA, each biological replicate was grouped in each treatment as expected for both species.

In terms of transcriptome assembly, contigs were merged to non-redundant unique transcripts as long as possible and clustered into unigenes. A total of 65,886 unigene contigs in ‘Goldrich’ and 78,215 in ‘Santa Rosa’ were finally filtered. Moreover, a minimum length of 200 bp was considered, while the maximum length reached was 13,689 and 14,450 for ‘Goldrich’ and ‘Santa Rosa’, respectively (Appendix A). Regarding unigene contig annotation, ‘Goldrich’ and ‘Santa Rosa’ merge assembly reached an overall annotated ratio of 61.22% (40,626) and 60.91% (39,067), respectively, which means that those contigs were annotated at least once in public databases (Table 4 and Appendix A). In addition, the NCBI (NT) was the public database that showed the greatest matches in both species (57.46% and 58.27%). Finally, around 44% of contigs corresponded to known proteins in both species, being functionally annotated 11,912 (18%) and 10,540 (16%) in apricot ‘Goldrich’ and Japanese plum ‘Santa Rosa’, respectively (Appendix A).

In the de novo transcriptome validation, the assemblies of ‘Goldrich’ and ‘Santa Rosa’ were aligned to *Prunus armeniaca* (‘Chuanzhihong’) and *Prunus salicina* (‘Zhongli’), respectively, reaching a high mapped reads coincidence of 88.44% and 97.96% (Table 4).

However, despite high alignment to reference genomes of both cultivars, after Cuffcompare analysis of unigene contigs (Table 5), ‘Goldrich’ showed a low exactly equal sequence (class code =) of 2.36% while ‘Santa Rosa’ obtained a higher ratio over 50 %. Furthermore, in the ‘Goldrich’ vs. ‘Chuanzhihong’ comparison, a higher possible pre-mRNA molecule (class code e) and exons falling into an intron of the reference (class code i) were found, unlike ‘Santa Rosa’ vs. ‘Zhongli’ who showed no sequences in the ‘e’ and ‘i’ classes.

### 2.3. Global Gene Expression, Gene Ontology, and Pathway Enrichment Analysis

According to the global gene expression analysis in ‘Goldrich’ and ‘Santa Rosa’ by treatment and treatment comparison, the 1-MCP treatment showed the highest number of expressed contigs (Figure 3A) for both species; meanwhile, in the 1-MCP vs. Ethrel comparison, gene induction prevailed over gene repression (Figure 3B).

Regarding gene ontology, all the processes involved at the biological level, cellular, and molecular function for each treatment comparison and species were shown (Appendix A). Regarding the biological level, metabolic processes and primary metabolism were the most abundant; cell, intracellular, and organelle were the most significant processes for cellular components; binding and catalytic activity were the main processes for molecular function.

According to the pathways involved in each treatment comparison and species, DEGs were linked to different metabolic pathways (Figure 4). Plant hormone signal transduction and photosynthesis were the most significant pathways for any treatment comparison, which is common in ‘Goldrich’ and ‘Santa Rosa’.

In the Venn diagrams, a total of 775 and 1607 DEGs functionally annotated in *Prunus persica* were filtered in ‘Goldrich’ and ‘Santa Rosa’ (Figure 5). In the diagrams, we can highlight a higher number of DEGs in ‘Santa Rosa’ as well as a major number of unique DEGs identified in the 1-MCP vs. Ethrel comparison in both species, which indicates that the opposite treatments generated more differences, as it was expected.

### 2.4. Differentially Gene Expression (DEG) Analysis between Treatments

Subsequently, after the filtering data, in order to select the most significant genes, we considered two criteria: (1) the highest FPKM difference and (2) the maximum log2FC between treatments. In apricot ‘Goldrich’, considering the highest FPKM differences, we found genes linked to pentose and glucuronate interconversions as pectinesterase (log2FC = 1.68) and polygalacturonase (log2FC = 1.65) which were upregulated in Ethrel, while auxin-responsive protein IAA (log2FC = 1.89) in the plant hormone signal transduction and ferredoxin (petF) (log2FC = 4.39) in the photosynthesis were upregulated by 1-MCP (Appendix A). However, if we consider the maximum log2FC as a criterion, chitinase [EC:3.2.1.14] (log2FC = 4.16) in the amino sugar and nucleotide sugar metabolism and 3-hydroxyisobutyryl-CoA hydrolase (HIBCH) (log2FC = 3.29) in the valine, leucine, and isoleucine degradation were the most significant, being upregulated by Ethrel. Regarding genes upregulated by 1-MCP, we can highlight those linked to photosynthesis as photosystem I subunit V (psaG) (log2FC = 6.44) and light-harvesting complex II chlorophyll a/b binding protein 4 (LHCB4) (log2FC = 5.74) (Appendix A).

In the case of Japanese plum ‘Santa Rosa’, considering the maximum FPKM difference, pectinesterase [EC:3.1.1.11] (log2FC = 3.56) in the pentose and glucuronate interconversions and flavonoid 3’-monooxygenase (CYP75B1) (log2FC = 3.82) in the flavonoid biosynthesis were upregulated by Ethrel, while aquaporin PIP (log2FC = 1.18) as a protein transporter and large subunit ribosomal protein LP2 (log2FC = 1.72) in the ribosome were upregulated by 1-MCP (Appendix A). Regarding the maximum log2FC, glutaredoxin 3 (grxC) (log2FC = 14.54) was linked to chaperones and folding catalysts, and palmitoyl-[glycerolipid] 7-desaturase (FAD5) (log2FC = 12.39) were upregulated in Ethrel, while photosystem I subunit V (psaG) (logFC = 6.26) in the photosynthesis and light-harvesting complex II chlorophyll a/b binding protein 4 (LHCB4) (log2FC = 5.47) in the photosynthesis antenna proteins were upregulated by 1-MCP (Appendix A). In addition, 1-aminocyclopropane-1-carboxylate synthase 1/2/6 (ACS1_2_6) in the cysteine and methionine metabolism linked to ethylene biosynthesis was upregulated by Ethrel treatment for both species.

### 2.5. Interaction Analysis of Proteins Related to DEGs

In terms of the protein–protein interaction network (PPI), we decided to consider common DEGs (−1 > log2FC > 1; *p*-value < 0.005) in both species for the most contrasting treatment comparison (1-MCP vs. Ethrel), including expressed genes linked within each network (Figure 6 and Appendix A). In the PPI, we can differentiate two main clusters where the predominant proteins are involved in photosynthesis (light green) and plant hormone signal transduction (light blue) (Figure 6).

In the photosynthesis cluster, LHCA2, LHCA3, LHCA4, LHCB1, LHCB2, LHCB5, and LHCB6 proteins are involved in photosynthetic carbon fixation and pentose and glucuronate interconversions, and although they showed low expression, all of them were upregulated by 1-MCP treatment. Regarding other proteins integrated into the photosynthetic network, psbR (photosystem II 10kDa protein) and psbP (photosystem II oxygen-evolving enhancer protein 2) were some of the most overexpressed protein-related genes by 1-MCP (Figure 6 and Figure 7, Appendix A).

The second cluster was mainly composed of plant hormone signal transduction proteins, EIN3, EIN4, ERF1, ETR2, or SNRK2.3, which interconnect this route with the MAPK signaling pathway, SNRK2.3 being interrelated with rbcS which in turn connects to carbon metabolism and photosynthetic carbon fixation (Figure 6). In the same cluster, EBF1_2 (EIN3-binding F-box protein) and PYL (abscisic acid receptor PYR/PYL family) were upregulated by Ethrel and 1-MCP treatment, respectively (Appendix A).

Moreover, a third tiny cluster was connected to photosynthesis and photosynthetic carbon fixation through petC (cytochrome b6-f complex iron-sulfur subunit), which was in turn upregulated by 1-MCP (Figure 6 and Figure 7, Appendix A). This third cluster contains proteins such as mMDH2, MDH, PMDH1, ACO, FDH, GAPCP-1, GAPC1, GAPDH, and pgm, which are key proteins involved in many pathways, specially phosphoglucomutase (pgm) which interconnects pathways such as amino sugar and nucleotide sugar metabolism, pentose and glucuronate interconversions or glycolysis, and gluconeogenesis.

### 2.6. qPCR Validation

Sixteen genes included in some of the main fruit ripening-related pathways were validated by qPCR, including pectinesterase (pentose and glucuronate interconversions), auxin-responsive protein (IAA, plant hormone signal transduction), and sucrose synthase (starch and sucrose metabolism) (Figure 8). Meanwhile, other specific genes linked to apricots, such as beta-carotene 3-hydroxylase (CRTZ, carotenoid biosynthesis)’ or flavonoid 3’-monooxygenase (CYP75B1) and chalcone synthase (CHS) in flavonoid biosynthesis were validated for Japanese plum ‘Santa Rosa’. qPCR results agree with the obtained RNA-Seq results in the assayed genes.

## 3. Discussion

### 3.1. Monitoring of Fruit Quality and Postharvest Traits

Over the years, ethylene inhibitors such as 1-MCP or ethylene precursors such as Ethrel have been extensively used in order to extend the fruit shelf-life period or accelerate fruit ripening in climacteric fruit, respectively [23]. These chemicals were satisfactorily applied in the present study to compare two opposite treatments linked to fruit ripening during the postharvest period in order to dissect this process at the gene expression level.

Overall, 1-MCP and Ethrel treatments in apricot and Japanese plum cultivars caused the expected effects. 1-MCP suppressed or delayed ethylene emission and lowered fruit softening rates, while Ethrel treatment produced higher softening rates and brought forward ethylene emission. However, after both treatments, the response presented by each cultivar was genotype-dependent, with apricot ‘Goldrich’ and Japanese plum ‘Santa Rosa’ being the most significant cultivars in terms of fruit firmness and ethylene emission. Furthermore, ‘Goldrich’ also displayed the most significant skin color changes between treatments, and ‘Santa Rosa’ was considered the most contrasting Japanese plum cultivar in terms of skin and flesh color between treatments. This result confirmed that critical ethylene emission is linked to skin and flesh color changes over time or between treatments [25]. Therefore, we can assert that these events, including fruit softening, ethylene emission, respiration rate, skin and flesh color changes, and chlorophyll breakdown, are all interrelated in a complex network, which should be studied more deeply at the molecular level.

### 3.2. De Novo Transcriptome Assembly, Functional Annotation, and Transcripts Classification

The sequencing results obtained a high Phred quality score (>95% in Q20 and >90% in Q30) in both species, which was considered high-quality sequencing [26]. The results showed thousands of expressed genes in both species; however, when they were blasted against public databases, only approximately 60% of unigenes (≈40,000 contigs) were annotated. Similar results were found in other *Prunus* species, such as almonds [27]. Moreover, around 44% of contigs corresponded to known proteins in both species, of which only approximately 18% and 16% in apricot and Japanese plum, respectively, have a known function. These low percentages clearly confirm a lack of genome annotation information concretely in stone fruit trees. In general, the 1-MCP treatment caused the highest number of expressed genes for both species. This trend was confirmed in the comparative transcript profiling of apricot [28], where three developmental stages were considered: immature green, mature firm ripe, and fully ripe. Furthermore, the 1-MCP vs. Ethrel comparison produced the maximum number of DEGs, as was expected.

On the other hand, de novo transcriptomes of ‘Goldrich’ and ‘Santa Rosa’ were aligned to *Prunus armeniaca* and *Prunus salicina* genomes, evidencing a high coincidence of mapped reads. However, ‘Goldrich’ showed a low number of exactly equal reads with apricot ‘Chuanzhihong’, while ‘Santa Rosa’ showed over 50% of equal sequences with the ‘Zhongli’ genome. These results evidence differences between ‘Goldrich’ and ‘Chuanzhihong’ cultivars in contrast to the high similarity of ‘Santa Rosa’ and ‘Zhongli’ genomes.

### 3.3. DEGs, PPI, and Related Pathways

Regarding the most significant genes involved in fruit ripening after 1-MCP and Ethrel treatments by the highest FPKM difference, in apricot, pectinesterase and polygalacturonase (pentose and glucuronate interconversions) were upregulated by Ethrel while auxin-responsive protein IAA (plant hormone signal transduction) and ferredoxin petF (photosynthesis) were upregulated by 1-MCP. On the other hand, in Japanese plum, pectinesterase and flavonoid 3’-monooxygenase CYP75B1 (flavonoid biosynthesis) were upregulated by Ethrel while aquaporin PIP (protein transporter LP2 in the ribosome) was upregulated by 1-MCP.

Other authors have previously reported polygalacturonase, pectin methylesterase, lyase, and rhamnogalacturonase as commonly degrading enzymes of pectins [29]. These enzymes have been extensively described in peach fruit and produce the depolymerization of pectins during the ripening process, where pectinesterase activity decreases and polygalacturonase increases [30]. In *Prunus salicina*, Lin et al. [31] confirmed a lower activity of cell wall-degrading enzymes such as pectinesterase, polygalacturonase, cellulase, and β-galactosidase after 1-MCP treatment. On the other hand, other works reported a notable increase in genes encoding IAA protein which were substantially upregulated at early ripening [28], as in our trial using 1-MCP. Regarding petF, this protein is considered a photosynthetic electron transporter such as PsbP, PsbR, and PsbY, which alongside photosynthetic-antenna proteins such as Lhca and Lhcb, is upregulated when we implement more efficient light use systems during fruit development as in the case of cherry [32]. In our assay, the delay of fruit ripening caused by 1-MCP maintained higher values of petF, similar to what occurs during cherry fruit development.

The flavonoid biosynthesis pathway has also proven a determinant pathway linked to the fruit ripening process in *Prunus* species [15]. In this work, a gene encoding the flavonoid 3’-monooxygenase enzyme increased as ripening progressed in different apricot genotypes. Furthermore, there are few works that evidence the role of aquaporins in the fruit ripening process. As is known, aquaporins are involved in water transport through the plasma membrane, and a recent study in sweet cherry asserted that they are downregulated towards maturity [33]. This result would be in accordance with the present assay, confirming higher expression of a gene encoding aquaporin PIP in 1-MCP treatment, which results in a less advanced ripening state.

On the other hand, if we compare genes encoding proteins and enzymes according to log2FC in apricot, chitinase (amino sugar and nucleotide sugar metabolism) and 3-hydroxyisobutyryl-CoA hydrolase HIBCH (valine, leucine, and isoleucine degradation) were the most significant being upregulated by Ethrel. Regarding Japanese plum, glutaredoxin 3, grxC (chaperones and folding catalysts) and palmitoyl-[glycerolipid] 7-desaturase FAD5 were both upregulated by Ethrel while psaG (photosynthesis, photosystem I subunit V) and LHCB4 (photosynthesis-antenna proteins, light-harvesting complex II chlorophyll a/b binding protein 4) were upregulated by 1-MCP.

Chitinases are PR proteins, and their regulation is controlled by abscisic acid (ABA), ethylene, jasmonic acid (JA), or salicylic acid (SA). Previous studies in bananas ensured that Ethylene exposure induced chitinase expression, occurring during fruit ripening, while 1-MCP treatment inhibited the expression [34]. In this assay endochitinase gene (MaECHI1) gradually increased during the ripening process. On the other hand, Acyl-CoA oxidases (ACXs) belong to the lipid metabolism and play a role in jasmonic acid (JA) biosynthesis, which provides protection against some pathogens. In previous studies in papaya, these *ACX* genes increase its transcript abundance, which supposes a defense system during fruit ripening [35], this was represented as a gene encoding HIBCH upregulated by Ethrel in our assay. Regarding glutaredoxins (GRXs), they are known as disulfide oxidoreductases and are involved in several biological processes. A recent work [36] suggested that GRXs interact with glutathione reductase, sulfiredoxin, or peroxiredoxin, favoring an antioxidative response in bananas.

In the case of *FAD* genes, they participate in the biosynthesis of esters through mediation in the accumulation of linoleic acid and linolenic acid as substrates. In a previous work studying apple, acetyl CoA, FAD2, FAD5, and LOX showed a higher expression in the latest fruit development stage. In addition, FAD2 and O-methyltransferase were overexpressed and correlated with ethylene emission [37].

Regarding photosynthesis related-proteins, psaG has the function of Lhca1/4 binding and regulating photosystem I in the *Arabidopsis thaliana* [38], while light-harvesting chlorophyll a/b-binding (Lhcb) is involved in the light-harvesting complex of photosystem II. In *Arabidopsis*, Waters et al. [39] assert that Lhcb photosynthetic genes could be target genes to golden2-like (GLK) transcription factors. These GLKs belong to the GARP family of MYB transcription factors, and they are linked to chloroplast development in plants [40]. More recently, Chen et al. [41] described *PpLhcb1.3*, *PpLhcb2.2*, and *PpLhca3* genes in peach, which showed a considerable decrease in concordance with the chlorophyll content during the fruit ripening process. Therefore, the least fruit ripening due to 1-MCP treatment produced an overexpression of genes encoding photosynthetic proteins such as psaG and Lhcb involved in photosystems I and II.

In addition, ACS1_2_6 in the cysteine and methionine metabolism was also among the most significant proteins encoded by genes from apricot and Japanese plum [42,43], being low in temperature and 1-MCP treatments being effective in delaying ethylene biosynthesis. As aforementioned, ACS is a key enzyme initiating ethylene biosynthesis and, in the current assay, was downregulated by 1-MCP and upregulated by Ethrel, which demonstrated the effectiveness of treatments applied for the control of ethylene-related genes, which in turn are deeply involved in ripening time and therefore affect the fruit shelf-life period, as shown in other studies [44,45].

Finally, the PPI network confirms the common interaction of some mentioned proteins in both species, highlighting photosynthesis and plant hormone signal transduction as major pathways. As we know, in climacteric fruit, ethylene is the main molecule regulating firmness, color change, and chlorophyll degradation. Moreover, the relationship between ethylene and auxin has also been studied. While ethylene accelerates the fruit ripening process, auxins inhibit it and are involved in fruit development [46].

In the case of the role of photosynthesis, over-ripening in other species, such as tomatoes, has been considered unimportant; however, it could be playing an important role in developing their seeds [47]. Conversely, in recent studies on tomatoes, photosynthetic proteins of PSI and PSII, including ferredoxin in the photosynthetic electron transport chain, remained higher in green fruit pericarp since this fruit remains photosynthetically active while these proteins decrease during the ripening process. On the other hand, aux/IAA plays an important role not only in fruit development but also in photosynthesis [48].

The presence of photosynthetic enzymes in the fruit ripening of *Prunus* species could be considered negligible considering that these fruits would no longer be photosynthetically active. However, the differences found between treatments are evident and cannot be underestimated. On the other hand, the fact that the expression of these genes is greater in the 1-MCP treatment does not mean that this is maintained over ripening time, but these genes are decreasing their expression during this period since the fruit has a negative photosynthetic balance. Therefore, one of the main reasons why 1-MCP might be favoring the expression of enzymes such as ferredoxin (PETF) in contrast to control and Ethrel treatment could be linked to the antioxidant regulation of compounds such as L-Ascorbic acid (Asc), which is the main water-soluble antioxidant [49,50]. L-ascorbate is synthesized by the Smirnoff–Wheeler (SW) pathway [51], where the GDP-L-galactose phosphorylase acts as a key enzyme [52]. In the gene validation of this study, we evaluated the expression level of genes encoding for GDP-L-galactose and ferredoxin, obtaining a greater expression in the 1-MCP treatment in both species, which would corroborate that 1-MCP applications could be indirectly favoring the antioxidant activity.

A similar approach in *Actinidia*, applying 1-MCP and Ethrel phytoregulators in green fleshy kiwi, was carried out by Salazar et al. [53], where after transcriptomic analysis, the main metabolic pathways were clustered, including pentose and glucuronate interconversions, citrate cycle, glycolysis and gluconeogenesis, starch and sucrose metabolism, porphyrin and chlorophyll, and photosynthesis in which photosynthesis-linked proteins in photosystem II delayed their repression due to 1-MCP application.

## 4. Materials and Methods

### 4.1. Plant Material, Experimental Design, and Evaluation of Fruit Quality and Postharvest

Apricot cultivars ‘Moniquí’, ‘Goldrich’, and ‘Dorada’, and Japanese plum cultivars ‘Golden Japan’, ‘Santa Rosa’, ‘Black Splendor’, and ‘Angeleno’ were collected at the CEBAS-CSIC experimental orchard located in Cieza-Calasparra (Murcia, Spain). The main differences and selection criteria among the selected apricot [7,54] and Japanese plum cultivars [55,56,57] were the high variability in terms of fruit quality and postharvest behavior (Table 1). Over 200 homogenous fruits were selected at commercial maturity per genotype and divided into three fruit sets (control, 1-MCP-treated, and Ethrel-treated). 1-MCP treatment (T1) was gasified as a SmartFresh™ product 0.14% (625 ppb) for 20 h while ethephon (T2) (Acid-2-chloroethyl-phosphonic) was applied by dipping the cultivars in buckets of distilled water as Ethrel^®^ 48 SL product (Ethrel) at 300 ppm during 15 min for apricots and 600 ppm during 20 min for plum cultivars. The control (T0), 1-MCP (T1), and Ethrel (T2) sets were stored in shelf-life standard conditions at 20 °C in order to monitor fruit quality and postharvest traits. Six fruit quality traits were evaluated, including fruit weight, over color, skin color, flesh color, soluble solids content, acidity at harvest, medium, and final period, according to the fruit maturity. Fruit weight was determined using a Blauscal digital balance (model AH-600), over color was evaluated visually on a scale from 1 to 4 (1, 0–25%; 2, 25–50%; 3, 50–75%; 4, 75–100%), skin and flesh color were measured using a Minolta Chroma Meter colorimeter (CR-300; Minolta, Ramsey, NJ, USA) using the CIELAB scale (h° = arctangent b*/a*) [58], soluble solids content was determined using a digital ATAGO^®^ (Tokyo, Japan) hand-held refractometer calibrated as the percentage of sucrose at 20 °C, and titratable acidity was determined by acid–base titration using 2 g of sample diluted in 30 mL of distilled water as malic acid grams per 100 mL [59]. Additionally, the total chlorophylls (a and b) and carotenoid content were determined by absorbance difference in the most contrasting apricot and plum cultivars, according to Nagata et al. [60]. Fruit samples were previously grained in liquid nitrogen and diluted in acetone:hexane (2:3) in a 1:20 *w*/*v* ratio, centrifuged at 3000× *g* for 10 min in a refrigerated centrifuge at 4 °C, and spectrophotometrically determined at 663, 645, 505, and 453 nm.

On the other hand, the most relevant traits directly linked to the fruit shelf-life period, including fruit softening, respiration rate, and ethylene emission, were monitored every other day. Firmness was quantified in Newtons (N) by a texture analyzer (TA.XT plus, Texture Technologies, Hamilton, MA, USA) using a 75 mm cylinder by compression of 5 mm depth. Regarding respiration rate and ethylene emission, the fruit was enclosed in jars of 750 mL over 2 h, and then respiration rate (g CO_2_ kg^−1^ h^−1^) was quantified by a CO_2_ analyzer (Thermox CG 1000, AMETEK^®^, EE.UU) while ethylene emission (mg kg^−1^ h^−1^) was determined by gas chromatography (ethylene chromatograph Konik^®^_,_ U.E). The experimental unit per cultivar, day, and treatment was 10 fruits except for respiration rate and ethylene emission, where a total of five fruits per treatment (T0, T1, and T2) were monitored. ANOVA analysis and Tukey multiple comparison tests (*p*-value ≤ 0.05) were applied in order to detect significant differences between treatments, and principal components analysis between treatments and cultivars was performed using INFOSTAT v16 software (Universidad Nacional de Córdoba, Argentina). PCA was constructed considering each trait as a variable and classifying by genotype (Appendix A) and by treatment, considering each genotype individually for both species (Appendix A).

### 4.2. RNA Isolation, Sequencing, and De Novo Assembly

Total RNA from 18 fruit samples, including three biological replicates from each treatment (T0, T1, and T2) and cultivar (‘Goldrich’ and ‘Santa Rosa’), were extracted. Approximately 200 mg of each fruit sample were grained in liquid nitrogen according to Le Provost et al.’s [61] protocol implemented for RNA extraction using CTAB buffer at 2%, Chloroform-isoamyl-alcohol 24:1, and LiCl 10 M. Finally, RNA samples were purified by the UltraClean Plant RNA Isolation Kit (Qiagen, Hilden, Germany) according to the manufacturer’s instructions. Quantity and quality of RNA were determined by spectrophotometer and agarose gel, and the higher quality samples were selected for sequencing. Library construction and sequencing were carried out by the Macrogen company (South Korea, Seoul) using the NovaSeq 6000 platform. Finally, 18 paired-end libraries with a fragment size of 151 bp were constructed using the TruSeq Stranded mRNA LT Sample prep kit, which resulted in approximately 40 to 60 million reads per library. The raw data are available in the sequence reads archive (SRA) with the bio project number PRJNA752255.

A quality control analysis of the sequenced raw reads was carried out. Overall reads quality, total bases, total reads, GC (%), and basic statistics were calculated (Appendix A). In order to reduce bias and artifacts such as low-quality reads, adaptor sequence, contaminant DNA, or PCR duplicates were removed. The quality of the produced data was determined by the Phred quality score at each cycle using FastQC. This software performs a quality check on the raw sequences before analysis to ensure data integrity. Its main function is importing BAM, SAM, FastQ files, and providing a quick overview of which section has problems. We used a Phred quality score of 20 which ensures 99% accuracy and reads over a score of 20 were accepted as good quality. A trimmomatic command line was used to remove adapter sequences and bases with base quality lower than three from the ends. The trimmed reads for all samples were merged into one file for transcriptome assembly. Merged data were assembled using the Trinity program, generally utilized for de novo reconstruction of transcriptomes. For assembled genes, the longest contigs of the assembly were filtered and clustered into non-redundant transcripts using the CD-HIT-EST program [62]. These transcripts were defined as unigenes and used for the subsequent annotation and ORF prediction. Then, for the functional annotation analysis, these unigenes were searched against several public databases such as NCBI Nucleotide (NT), NCBI Non-Redundant Protein (NR), Pfam, Gene Ontology (GO), UniProt, and EggNOG using BLASTN of NCBI BLAST and BLASTX of DIAMOND software v. 0.9.24 [63].

### 4.3. Gene Expression Profile, Gene Ontology, and Network Analysis

The abundance of unigenes across samples was estimated by the RSEM algorithm, and the expression level was calculated as the read count and fragments per kilobase of exon model per million reads mapped (FPKM) (Appendix A). DEGs analysis was carried out using the R Bioconductor package edgeR for all possible comparisons between treatments (T0 vs. T1, T0 vs. T2, and T1 vs. T2) and genotypes (‘Goldrich’ and ‘Santa Rosa’) (Appendix A). Regarding gene enrichment overview, we used GO terms for each biological process using BiNGO version 3.0.4 tool (Biological Networks Gene Ontology). Cytoscape version 3.8.0 [64] was implemented in order to calculate overrepresented GO terms and display those significant in a network. Furthermore, the ClueGO version 2.5.8 [65,66] plugin of Cytoscape version 3.8.0 was used for enrichment analysis of DEGs [67]. The GO terms were classified as molecular function, biological process, or cellular component (Appendix A). Additionally, the Kyoto Encyclopedia of Genes and Genome (KEGG) public database was used for the analysis of metabolic and regulatory pathways. Functional classes of DEGs were determined, showing significant GO terms and pathways, based on hypergeometric distribution. A corrected *p*-value ≤ 0.05 according to the Benjamini–Hochberg method [68] was implemented to display gene enrichment (Appendix A). A protein–protein interaction network (PPI) was built using protein IDs of DEGs (−1 > log2 FC > 1; *p*-value < 0.005) from *Prunus persica* as a comparative model and the STRING version 1.7.5 database [69], considering a network type full chain, confidence (score) limit 0.6, no additional interactors, and using smart delimiters. The DEGs were grouped based on two parameters, stringdb text mining and the metabolic pathways, obtained in previous analyses. Finally, Cytoscape version 3.8.0 software was used in order to visualize the PPI network to identify the key proteins in the regulatory pathways (Appendix A). In the de novo assembly validation, GMAP [70] was used to align assembled transcripts of ‘Goldrich’ and ‘Santa Rosa’ to the reference genomes *Prunus armeniaca* v1.0 (‘Chuanzhihong’) [71] and *Prunus salicina* v1.0 (‘Zhongli’) [72], respectively. In addition, to analyze the transcribed fragments (transfrag) and to define the different classes of transcripts, Cufflinks cuffcompare v2.2.1 was applied (Appendix A).

### 4.4. RT-qPCR Validation

RT-qPCR validation was carried out using three cDNA biological and two technical replicates from a fruit sample pull of each treatment and species. A total of 16 genes were validated, of which 8 were for apricot ‘Goldrich’ and 8 were for Japanese plum ‘Santa Rosa’ (Appendix A). The oligonucleotide primers were designed using Primer3 version 0.4.0, for apricot ‘Goldrich’: polygalacturonase (PG), pectinesterase (PME-GD), auxin-responsive protein (IAA-GD), beta-carotene 3-hydroxylase (CRTZ), GDP-L-galactose phosphorylase (VTC2), UDP-glucuronate decarboxylase (UXS1), 3-hydroxyisobutyryl-CoA hydrolase (HIBCH), and sucrose synthase (SUSY-GD); for Japanese plum ‘Santa Rosa’ primers: auxin-responsive protein (IAA-SNR), pectinesterase (PME-SNR), 1-aminocyclopropane-1-carboxylate synthase (ACS), ferredoxin (PETF), flavonoid 3’-monooxygenase (CYP75B1), chalcone synthase (CHS), myo-inositol-1-phosphate synthase (INO1), and sucrose synthase (SUSY-SNR). PME, IAA, and SUSY were exclusively designed using sequences of ‘Goldrich’ and ‘Santa Rosa’ independently. Several housekeeping reference genes were assayed using two internal controls: RNA polymerase II (RPII) and Ubiquitin 10 (UBQ10) in both species [73]. The cDNA was synthesized using SuperScript III Reverse Transcriptase (Thermo Fisher Scientific), and RT-qPCR experiments were conducted in the One Step Plus real-time PCR system (Applied Biosystems). For all RT-qPCR reactions, a 10 μL mix was made, including 5 μL Power SYBR^®^ Green PCR Master Mix (Applied Biosystems), 0.5 μL of each primer (5 μM), and 2 µL of cDNA (2.5 ng/µL). Amplification conditions were 10 min at 95 °C, followed by 40 cycles of 15 s at 95 °C, and 1 min at 60 °C, and for melting curves, 15 s at 95 °C and 1 min at 60 °C, increasing by 0.3 °C until 0.15 s at 95 °C. Normalized Relative Quantification (NRQ) for gene expression levels was carried out using the modified Pfaffl method [74] based on the expression levels of the target gene vs. a housekeeping gene.

## 5. Conclusions

Whole transcriptome analysis of apricot and Japanese plum fruits after 1-MCP (ethylene-inhibitor) and Ethrel (ethylene-precursor) treatments revealed the significance of photosynthesis and plant hormone signal transduction-related genes on fruit ripening. The effect of ethylene regulator treatments was clearly genotype-dependent and determined by the genetic background of each cultivar. De novo transcriptome analysis of apricot ‘Goldrich’ and Japanese plum ‘Santa Rosa’ allowed the identification of thousands of DEGs influenced by these regulators. However, there is still a widespread lack of knowledge due to the high number of non-annotated genes. The metabolic pathways that encompass the most important differences in fruit ripening are plant hormone signal transduction and photosynthesis, which interact with pathways such as pentose and glucuronate interconversions, glycolysis and gluconeogenesis, starch and sucrose metabolism, cysteine and methionine, and porphyrin and chlorophyll metabolism. Within this framework, gene encoding pectinesterase was the most determinant in both species, as well as polygalacturonase and ACS, which were upregulated by Ethrel. However, new encoding key genes such as aquaporin PIP (protein transporter LP2 in the ribosome), chitinase (amino sugar and nucleotide sugar metabolism), glutaredoxin 3 (chaperones and folding catalysts), and psaG (photosynthesis) showed particular relevance. Genes linked to photosynthesis mostly delayed their repression due to 1-MCP application. Additionally, the over-expression of genes encoding GDP-L-galactose and ferredoxin in the ascorbate and aldarate metabolism and photosynthesis pathways caused by 1-MCP increased the antioxidant activity and therefore slowed down the fruit senescence process.

## Figures and Tables

**Figure 1 ijms-23-11045-f001:**
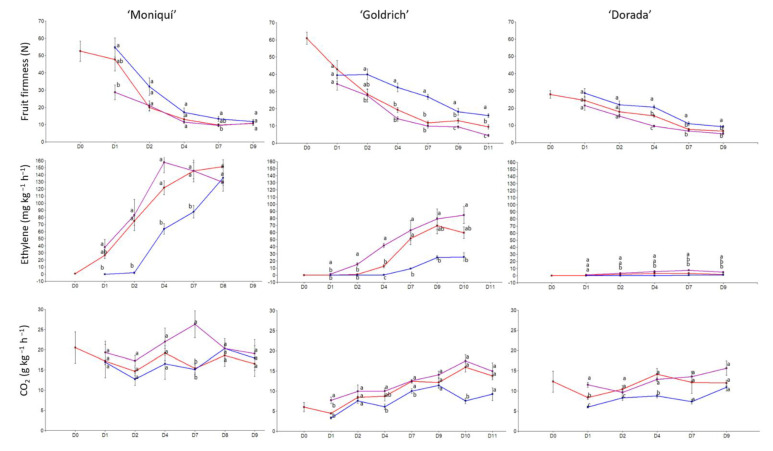
Evolution of fruit firmness by compression (N), ethylene emission (mg kg^−1^ h^−1^) and CO_2_ emission (g kg^−1^ h^−1^) of apricot cultivars assayed during the shelf-life period at 20 °C for control (red color), 1-MCP (blue color), and Ethrel treatments (purple color). Letters show significant differences in the Tukey test (*p*-value < 0.05).

**Figure 2 ijms-23-11045-f002:**
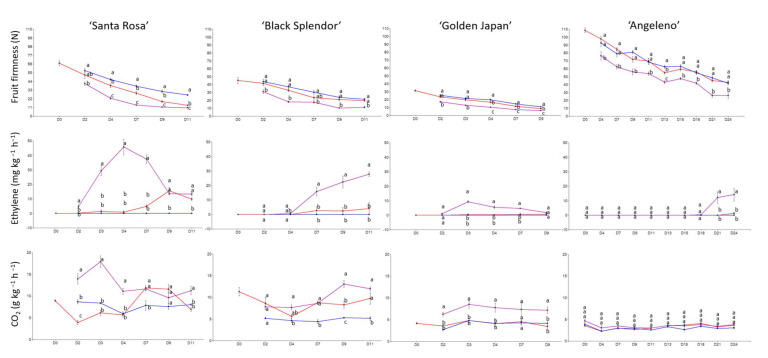
Evolution of fruit firmness by compression (N), ethylene emission (mg kg^−1^ h^−1^) and CO_2_ emission (g kg^−1^ h^−1^) of Japanese plum cultivars assayed during the shelf-life period at 20 °C for control (red color), 1-MCP (blue color), and Ethrel treatments (purple color). Letters show significant differences in the Tukey test (*p*-value < 0.05).

**Figure 3 ijms-23-11045-f003:**
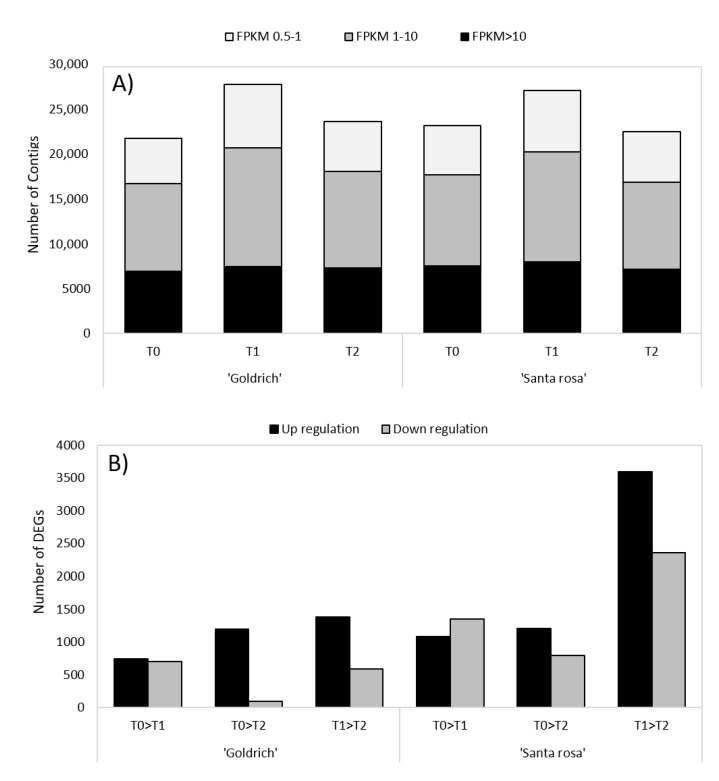
(**A**) Summary of the global gene expression in apricot ‘Goldrich’ and Japanese plum ‘Santa Rosa’ for different treatments (control, T0; 1-MCP, T1; Ethrel, T2) and (**B**) the number of up and down regulation DEGs (−1 > log2FC > 1; *p*-value < 0.005) for each treatment comparison (T0 vs. T1, T0 vs. T2, and T1 vs. T2).

**Figure 4 ijms-23-11045-f004:**
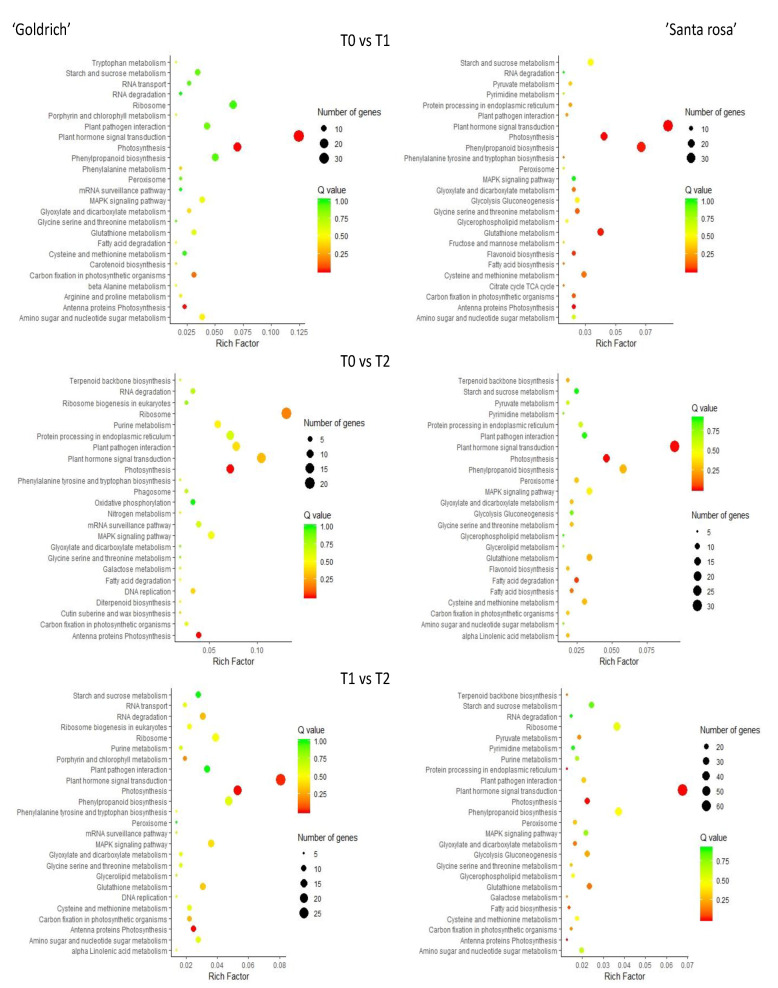
Metabolic pathway analysis based on DEGs filtering (−1 > log2FC > 1; *p*-value < 0.005) for T0 vs. T1, T0 vs. T2, and T1 vs. T2 comparison in apricot ‘Goldrich’ and Japanese plum ‘Santa Rosa’. Advanced bubble chart shows enrichment of DEGs to each signaling pathway. The *Y*-axis label represents the pathway, and the *X*-axis label represents the rich factor. The size and color of the bubble represent the amount of DEGs enriched in pathway and enrichment significance, respectively.

**Figure 5 ijms-23-11045-f005:**
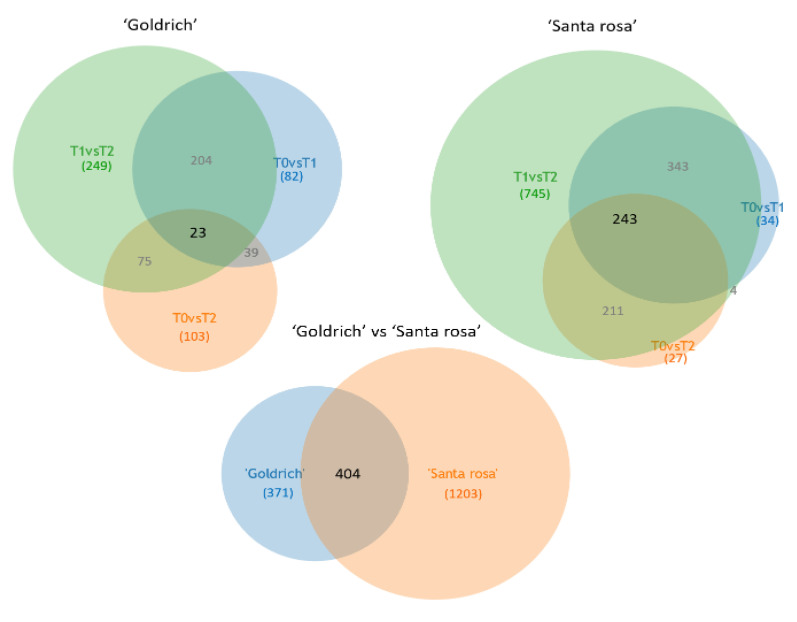
Venn diagrams of genes functionally annotated between treatment comparisons (T0 vs. T1, T0 vs. T1, and T1 vs. T2) and between apricot ‘Goldrich’ and Japanese plum ‘Santa Rosa’ after filtering (DEGs, −1 > log2FC > 1; *p*-value < 0.005).

**Figure 6 ijms-23-11045-f006:**
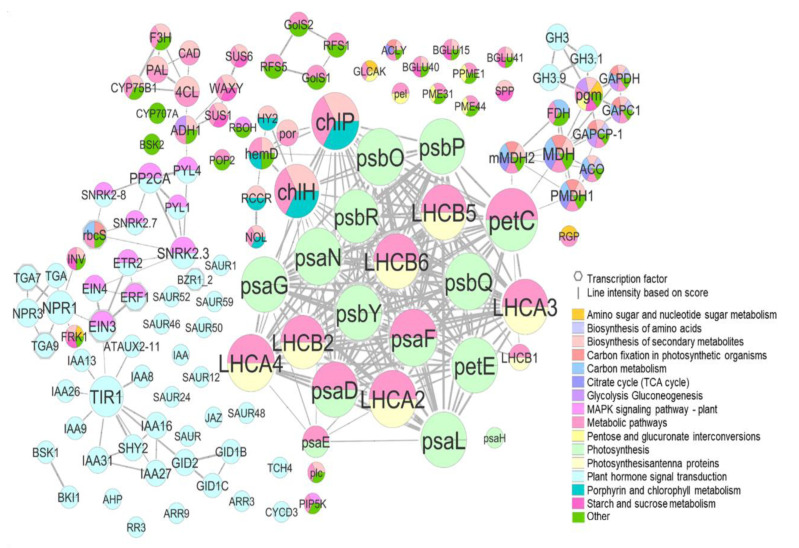
Protein–protein interaction network (PPI) of differentially expressed genes including the most DEGs common in both species (DEGs, −1 > log2FC > 1; *p*-value < 0.005) and expressed genes interconnected between them in T1 vs. T2 comparison. The node represents the protein, the size of the node is proportional to the number of proteins that interact, and the interaction with a score >0.6 were chosen for analysis.

**Figure 7 ijms-23-11045-f007:**
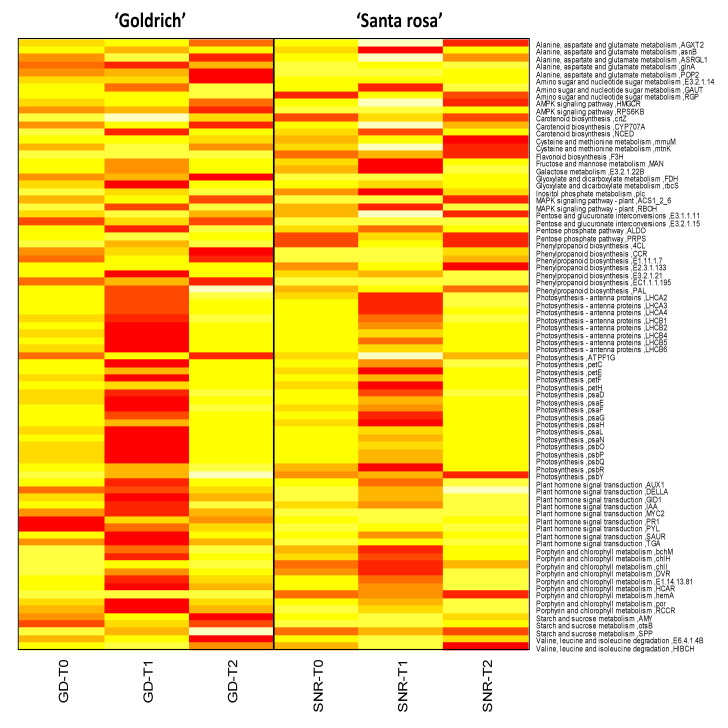
Gene expression of the most significant genes (DEGs, −1 > log2FC > 1; *p*-value < 0.005) and its homologous protein common in both species *Prunus armeniaca* (‘Goldrich’) and *Prunus salicina* (‘Santa Rosa’) for each treatment (control, T0; 1-MCP, T1; Ethrel, T2). Red indicates a maximum value of fragments per kilobase of exon model per million reads mapped (FPKM), while light yellow indicates a minimum FPKM.

**Figure 8 ijms-23-11045-f008:**
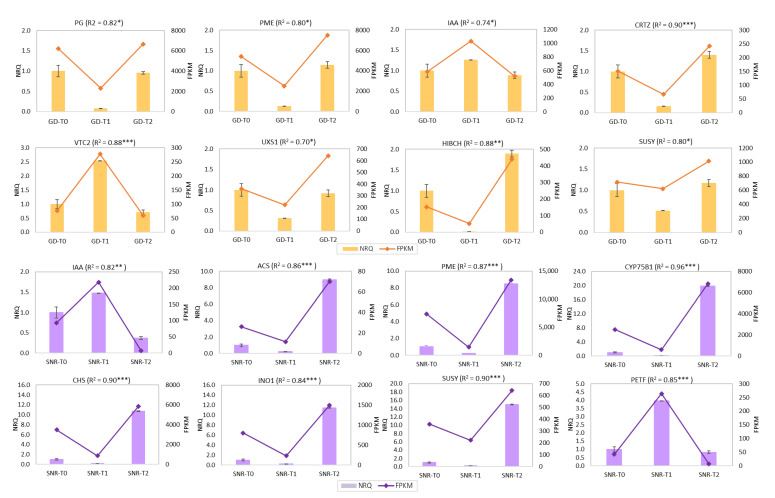
RNAseq validation using NRQ (Normalized Relative Quantification) by RT-qPCR and FPKM (Fragment per Kilobase Million) of 16 genes using three biological and two technical replicates. The orange color shows 8 genes validated in apricot ‘Goldrich’: polygalacturonase (PG), pectinesterase (PME), auxin-responsive protein (IAA), beta-carotene 3-hydroxylase (CRTZ), GDP-L-galactose phosphorylase (VTC2_5), UDP-glucuronate decarboxylase (UXS1), 3-hydroxyisobutyryl-coa hydrolase (HIBCH), and sucrose synthase (SUSY); while purple color shows 8 genes validated in Japanese plum ‘Santa Rosa’: auxin-responsive protein (IAA), pectinesterase (PECT), 1-aminocyclopropane-1-carboxylate synthase (ACS), ferredoxin (PET’), flavonoid 3’-monooxygenase (CYP75B1), chalcone synthase (CHS), inositol phosphate metabolism (INO1), and sucrose synthase (SUSY). In brackets, Pearson correlation is shown (*p*-value < * 0.05, ** 0.01, *** 0.005). PME, IAA, and SUSY were exclusively designed using sequences of ‘Goldrich’ and ‘Santa Rosa’ independently.

**Table 1 ijms-23-11045-t001:** Description of apricot and Japanese plum cultivars assayed, including the pedigree, origin, and main agronomical traits.

Variety	Pedigree	Origin	Self-Compatibility	Flowering	Ripening	Skin Color	Flesh Color	Postharvest
Moniquí	Unknown	Spain	Self-compatible	Late	Intermediate	White–pink	White	Fast
Dorada	Bergeron × Moniquí	Spain	Self-compatible	Late	Late	Light orange	Yellow	Intermediate
Goldrich	Sunglo × Perfection	USA	Self-incompatible	Late	Intermediate	Orange	Intense orange	Slow
Golden Japan	(*P. simonii* × *P. salicina*) × (*P.cerasifera* × *P. munsoniana*)	USA	Self-incompatible	Late	Early	Golden yellow	Pale yellow	Intermediate
Black Splendor	Unknown	USA	Self-incompatible	Early	Early	Blue–black	Red	Intermediate
Santa Rosa	Unknown	USA	Self-compatible	Early	Early	Red	Yellow–red	Fast
Angeleno	Unknown	USA	Self-incompatible	Late	Late	Purple–black	Yellow	Slow

**Table 2 ijms-23-11045-t002:** Descriptive statistic of fruit quality traits in apricot (*Prunus armeniaca* L.) cultivars at harvest time. FW: fruit weight (g); OVC: over color (1–4); SKC: skin color (CIE L*, a*, b*, C, and h°); FLSC: flesh color (CIE L*, a*, b*, C, and h°); SSC: soluble solid content (%); MALIC: malic acid content (g 100 mL^−1^); SSC/MALIC: soluble solid and malic acid content ratio.

Trait	‘Moniquí’				‘Goldrich’				‘Dorada’			
	Average	SD	Min	Max	Average	SD	Min	Max	Average	SD	Min	Max
FW	68.62	9.22	56.10	84.13	92.35	10.88	77.57	108.47	79.19	7.22	68.37	87.18
OVC	1.13	0.83	0.00	2.00	0.30	0.48	0.00	1.00	0.30	0.48	0.00	1.00
SKC (L*)	71.03	2.31	68.63	75.33	62.94	1.02	61.00	64.39	72.27	1.27	69.92	74.04
SKC (a*)	−4.12	2.68	−7.47	0.20	10.73	2.96	6.93	14.62	9.57	3.10	5.54	15.41
SKC (b*)	43.15	3.08	38.88	48.89	48.27	2.16	44.77	50.97	50.90	1.35	48.90	53.56
SKC (C)	43.49	3.02	39.13	49.18	49.54	2.38	45.36	52.61	51.92	1.42	49.88	54.70
SKC (h°)	95.46	3.65	89.55	100.39	77.50	3.21	73.64	82.24	79.37	3.37	73.26	83.92
FLSC (L*)	69.53	3.04	63.64	74.35	63.79	1.02	62.36	65.42	69.36	2.64	63.02	72.59
FLSC (a*)	−2.21	1.11	−3.49	−0.48	18.39	1.25	16.16	19.70	6.71	1.46	4.88	9.29
FLSC (b*)	26.90	1.47	24.68	29.06	49.57	1.21	47.10	50.89	48.09	1.67	45.41	51.27
FLSC (C)	27.02	1.44	24.93	29.22	52.89	1.21	50.45	54.51	48.58	1.76	45.67	51.65
FLSC (h°)	94.79	2.44	90.96	98.17	69.63	1.33	68.17	72.32	82.11	1.57	79.31	83.86
SSC	14.50	0.26	14.30	14.80	12.57	0.93	11.80	13.60	13.13	0.21	12.90	13.30
MALIC	1.64	0.02	1.62	1.65	3.10	0.04	3.05	3.13	1.06	0.10	0.96	1.15
SSC/MALIC	8.86	0.18	8.67	9.02	4.05	0.28	3.78	4.35	12.42	1.17	11.57	13.75

**Table 3 ijms-23-11045-t003:** Descriptive statistic of fruit quality traits in Japanese plum cultivars at harvest time. FW: fruit weight (g); OVC: over color (1–4); SKC: skin color (CIE L*, a*, b*, C and h°); FLSC: flesh color (CIE L*, a*, b*, C and h°); SSC: soluble solid content (%); MALIC: malic acid content (g 100 mL^−1^); SSC/MALIC: soluble solid and malic acid content ratio.

Trait	‘Santa Rosa’				‘Black Splendor’				‘Golden Japan’				‘Angeleno’			
	Average	SD	Min	Max	Average	SD	Min	Max	Average	SD	Min	Max	Average	SD	Min	Max
FW	55.88	8.29	46.32	73.99	82.60	12.97	63.47	104.49	45.08	7.30	35.98	63.71	70.39	8.74	59.86	86.00
OVC	4.00	0.00	4.00	4.00	4.00	0.00	4.00	4.00	4.00	0.00	4.00	4.00	4.00	0.00	4.00	4.00
SKC (L*)	34.28	1.77	31.65	37.09	28.69	1.95	26.46	31.97	60.36	1.21	57.85	62.79	35.95	3.22	32.15	41.67
SKC (a*)	25.31	2.76	20.85	29.40	8.80	2.77	4.75	14.02	−6.79	0.78	−7.81	−5.50	19.81	3.45	14.43	26.03
SKC (b*)	8.46	1.89	5.65	11.06	1.16	1.36	−0.98	3.97	38.74	2.10	35.61	42.21	5.76	3.55	2.52	14.48
SKC (C)	26.71	3.20	21.60	31.42	9.04	2.90	5.13	14.58	39.37	2.03	36.31	42.64	20.80	4.34	14.68	29.78
SKC (h°)	18.23	2.11	15.15	21.42	11.11	5.88	2.76	21.09	99.97	1.34	97.97	101.68	15.09	5.78	8.83	29.08
FLSC (L*)	52.91	3.48	46.87	58.73	33.45	2.35	28.70	37.26	54.59	2.43	49.52	57.91	53.81	1.22	51.95	56.28
FLSC (a*)	15.87	7.58	0.62	24.88	28.50	2.27	24.47	33.05	−4.61	0.70	−5.90	−3.38	−7.32	0.74	−8.33	−6.12
FLSC (b*)	28.70	3.40	24.12	33.77	13.01	1.67	9.73	15.69	31.39	1.99	27.67	34.33	36.41	1.59	33.62	39.32
FLSC (C)	33.82	1.25	32.47	36.01	31.37	2.65	26.34	36.25	31.75	1.99	27.97	34.61	37.15	1.55	34.36	39.82
FLSC (h°)	61.27	14.17	44.05	88.89	24.37	1.65	21.58	27.68	98.32	1.27	96.54	100.69	101.38	1.23	99.13	102.86
SSC	13.80	0.92	13.00	14.80	13.80	0.26	13.50	14.00	10.30	0.17	10.20	10.50	17.40	1.13	16.10	18.10
MALIC	2.35	0.02	2.33	2.37	1.72	0.08	1.63	1.77	2.02	0.05	1.97	2.06	1.16	0.11	1.03	1.23
SSC/MALIC	5.88	0.34	5.58	6.24	8.04	0.44	7.67	8.53	5.09	0.09	5.18	5.00	15.03	0.53	14.63	15.63

**Table 4 ijms-23-11045-t004:** Summary of statistics of de novo transcriptome assembly and gene annotation. De novo assembly was performed by Trinity software following the standard manual, and functional annotation of unigenes was searched against Kyoto Encyclopedia of Genes and Genomes (KEGG), NCBI Nucleotide (NT), Pfam, Gene ontology (GO), NCBI non-redundant Protein (NR), UniProt, and EggNOG using BLASTN of NCBI BLAST and BLASTX of DIAMOND software with an E-value default cutoff of 1.0 × 10^−5^. N50 is the shortest length sequence at 50% of the assembled transcript. Mapping statistics of reads to the transcriptome assemblies and peach genome are also provided, which were obtained using the Bowtie v2 package. * *Prunus armeniaca* genome (‘Chuanzhihong’) ** *Prunus salicina* genome (‘Zhongli’).

Statistics	‘Goldrich’	‘Santa Rosa’
Unigene contigs	65,886	78,215
Min. Lenght (bp)	201	201
Max. Lenght (bp)	13,689	14,450
N50	1204	1256
Annotation by Blast (%)	61.22	60.91
Reads mapped to reference genome (%)	88.44 *	97.96 **

**Table 5 ijms-23-11045-t005:** Summary of the transcript class assembled in apricot ‘Goldrich’ and Japanese plum ‘Santa rosa’ using Cufflinks cuffcompare v2.2.1. Class codes described by Cuffcompare: =, exactly equal to the reference annotation; C, contained in the reference annotation; E, possible pre-Mrna molecules; I, an exon falling into an intron of the reference; O, unknown generic overlap with reference; P, possible polymerase run-on; U, unknown intergenic transcript; X, antisense isoform.

Class Code	‘Goldrich’		‘Santa Rosa’	
	No. of Transcripts	%	No. of Transcripts	%
	1554	2.36	12,192	54.10
C	4558	6.92	33,465	21.41
E	33,310	50.56	-	-
I	8257	12.53	-	-
O	3032	4.60	4535	5.60
P	2647	4.02	4655	1.92
U	7722	11.72	17,155	10.09
X	4806	7.29	6213	6.87
Total	65,886	100	78,215	100

## Data Availability

The raw reads from the Illumina NovaSeq 6000 platform were deposited into NCBI in the sequence reads archive (SRA) under the BioProject accession number PRJNA752255.

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
