# Peer review of "Whole Transcriptome Analyses of Apricots and Japanese Plum Fruits after 1-MCP (Ethylene-Inhibitor) and Ethrel (Ethylene-Precursor) Treatments Reveal New Insights into the Physiology of the Ripening Process"

_ijms, 2022, doi:10.3390/ijms231911045_

Round 1
Reviewer 1 Report
The study is important to understand the physiology of ripening process. The research article is written well.
Author Response
Dear Ms. Pari Xu, Assistant Editor of International Journal of Molecular Sciences,
please find enclosed the manuscript ijms-1916500-R1 entitled “Whole transcriptome analysis of apricot and Japanese plum fruits after 1-MCP (ethylene-inhibitor) and Ethrel (ethylene-precursor) treatments revealed new insights into the physiology of ripening process” which is the revised version of the manuscript which we would like to publish in your journal.
According with the suggestions of the Reviewer 1 we have revised the manuscript incorporating the proposed revisions indicating these revisions with the control of changes of the WORD document.
We deeply appreciate the efforts of the reviewer in the improvement of the manuscript for a future publication.
Regarding reviewer's 1 comments (R1):
R1: The study is important to understand the physiology of ripening process. The research article is written well.
Authors: We agree and thank the Reviewer 1 for their comments about the revision of this work and for considering this manuscript suitable for future publication. In addition, all the suggestions and revisions of the reviewer have been incorporated indicating these revisions with the “Track Changes” of the WORD document.
In the new version, we have improved the Introduction section talking about the family and the ploidy level of these species (lines 41-44), as well as we make reference to the role of the ethylene in the ripening process (81-92). In addition, new information was added in the Discussion section (451-457), highlighting the role of ACS in ethylene biosynthesis and the effectiveness of the treatments. Regarding Materials and Methods section, some additional information about functional annotation was added (578-583).
Finally, English language and style were checked and improved.
We deeply appreciate the efforts of the reviewer in the improvement of the manuscript for a future publication.
Yours faithfully,

Reviewer 2 Report
In this manuscript, the authors first did statistic analyses of fruit traits of apricot and Japanese plum cultivars and then carried out the RNAseq assay of apricot and Japanese plum upon ethylene inhibitor and precursor treatment. Through analyzing the transcriptome data, the authors showed that photosynthesis and hormonal signaling transduction pathways are significantly changed upon treatments. Q-PCR was applied to test candidate genes involved in fruit ripening. My comments are listed in detail as follows.
1. A brief information of apricot and plum should be provided in the Introduction, for example, do they belong to the same family or genus?
2. How many chromosomes in their genomes? Are they diploid or polyploid?
3. Line 148 and Line 207, should it be evolution or evaluation?
4. Is the PCA analysis carried out in the RNAseq analysis?
5. In the Results section, the title of each sub-section should be a brief conclusion of the results below.
6. Figure 7, typically, red color indicates maximum.
7. As ethylene inhibitor and precursor, more information of ethylene's role in fruit ripening should be introduced or discussed in the introduction and discussion.
Author Response
Dear Ms. Pari Xu, Assistant Editor of International Journal of Molecular Sciences,
please find enclosed the manuscript ijms-1916500-R1 entitled “Whole transcriptome analysis of apricot and Japanese plum fruits after 1-MCP (ethylene-inhibitor) and Ethrel (ethylene-precursor) treatments revealed new insights into the physiology of ripening process” which is the revised version of the manuscript which we would like to publish in your journal.
According with the suggestions of the Reviewer 2 we have revised the manuscript incorporating the proposed revisions indicating these revisions with the control of changes of the WORD document.
We deeply appreciate the efforts of the reviewer in the improvement of the manuscript for a future publication.
Regarding reviewer's 2 comments (R2):
R2: In this manuscript, the authors first did statistic analyses of fruit traits of apricot and Japanese plum cultivars and then carried out the RNAseq assay of apricot and Japanese plum upon ethylene inhibitor and precursor treatment. Through analyzing the transcriptome data, the authors showed that photosynthesis and hormonal signaling transduction pathways are significantly changed upon treatments. Q-PCR was applied to test candidate genes involved in fruit ripening. My comments are listed in detail as follows.
Authors: We agree and thank the Reviewer 2 for their valuable comments about the revision of this work. In addition, all the suggestions and revisions of the reviewer have been incorporated indicating these revisions with the “Track Changes” of the WORD document.
R2: A brief information of apricot and plum should be provided in the Introduction, for example, do they belong to the same family or genus?
Authors: As the reviewer suggest, new information about family and genus of these species was added in the Introduction section in order to help readers place the context in terms of taxonomy and their kinship degree (lines 41-44).
R2: How many chromosomes in their genomes? Are they diploid or polyploid?
Authors: According to reviewer comment, we would like to add that these species are diploid and both comprise eight chromosomes. This information, including ploidy level and the chromosomal endowment was added in the Introduction section (line 43).
R2: Line 148 and Line 207, should it be evolution or evaluation?
Authors: thank you for comment. In these lines, we make reference to “evolution” better than “evaluation” because we talk about traits measured over a timeline (different days).
R2: Is the PCA analysis carried out in the RNAseq analysis?
Authors: Yes, it is, we used all of transcripts and the biological replicates from the de novo transcriptome (supplementary figure S7).
R2: In the Results section, the title of each sub-section should be a brief conclusion of the results below.
Authors: The title of each sub-section in the Results section has been revised according to the suggestions of the reviewer.
R2: Figure 7, typically, red color indicates maximum.
Authors: We agree and thank you for your comment. Really, in the legend description there was a mistake. Therefore, red color corresponds to maximum and white-yellow color to minimum. The description legend was modified.
R2: As ethylene inhibitor and precursor, more information of ethylene's role in fruit ripening should be introduced or discussed in the introduction and discussion.
Authors: As reviewer suggests additional information was added making reference to the role of the ethylene in the ripening process (81-92), as well as new information was added in the Discussion section (451-457), highlighting the role of ACS in ethylene biosynthesis and the effectiveness of the treatments.
We deeply appreciate the efforts of the reviewer in the improvement of the manuscript for a future publication.
Yours faithfully,

Reviewer 3 Report
In this research paper, author have performed whole transcriptome analysis of apricot and Japanese plum fruits after 1-MCP (ethylene-inhibitor) and Ethrel (ethylene-precursor) treatments to check the new insight and to check the effect on ripening process. They have performed high-throughput sequencing analysis (RNA-Seq) to understand physiology of Prunus fruit ripening. Authors have used precise method to perform transcriptome analysis. Therefore, this reviewer would like to suggest that the manuscript is acceptable for publication in the journal. Some references cited in the manuscript should be updated to the recently published papers and arrange the bibliography accordingly.
Author Response
Dear Ms. Pari Xu, Assistant Editor of International Journal of Molecular Sciences,
please find enclosed the manuscript ijms-1916500-R1 entitled “Whole transcriptome analysis of apricot and Japanese plum fruits after 1-MCP (ethylene-inhibitor) and Ethrel (ethylene-precursor) treatments revealed new insights into the physiology of ripening process” which is the revised version of the manuscript which we would like to publish in your journal.
According with the suggestions of the Reviewer 3 we have revised the manuscript incorporating the proposed revisions indicating these revisions with the control of changes of the WORD document.
We deeply appreciate the efforts of the reviewer in the improvement of the manuscript for a future publication.
Regarding reviewer's 3 comments (R3):
R3: In this research paper, author have performed whole transcriptome analysis of apricot and Japanese plum fruits after 1-MCP (ethylene-inhibitor) and Ethrel (ethylene-precursor) treatments to check the new insight and to check the effect on ripening process. They have performed high-throughput sequencing analysis (RNA-Seq) to understand physiology of Prunus fruit ripening. Authors have used precise method to perform transcriptome analysis. Therefore, this reviewer would like to suggest that the manuscript is acceptable for publication in the journal. Some references cited in the manuscript should be updated to the recently published papers and arrange the bibliography accordingly.
Authors: We agree and thank the Reviewer 3 for their comments about the revision of this work and for suggesting this manuscript as acceptable for future publication. In addition, all the suggestions and revisions of the reviewer have been incorporated indicating these revisions with the “Track Changes” of the WORD document.
In the new version, we have improved the Introduction section talking about the family and the ploidy level of these species (lines 41-44), as well as we make reference to the role of the ethylene in the ripening process (81-92). In addition, new information was added in the Discussion section (451-457), highlighting the role of ACS in ethylene biosynthesis and the effectiveness of the treatments. Regarding Materials and Methods section, some additional information about functional annotation was added (578-583).
Finally, new references were added in the introduction (lines 83 [21] and 91 [22]) and discussion (lines 453-458, [44,45]) sections, as well as English language and style were checked and improved.
We deeply appreciate the efforts of the reviewer in the improvement of the manuscript for a future publication.
Yours faithfully,

Round 2
Reviewer 2 Report
No further comments